# Representation Learning on Biomolecular Structures using Equivariant Graph Attention

**Tuan Le**
Bayer AG
Freie Universität Berlin
tuan.le2@bayer.com

**Frank Noé**
Microsoft Research AI4Science
Freie Universität Berlin
franknoe@microsoft.com

**Djork-Arné Clevert**[*]
Bayer AG
djork-arne.clevert@pfizer.com

## Abstract

Learning and reasoning about 3D molecular structures with varying size is an emerging and important challenge in machine learning and especially in the development of biotherapeutics. Equivariant Graph Neural Networks (GNNs) can simultaneously leverage the geometric and relational detail of the problem domain and are known to learn expressive representations through the propagation of information between nodes leveraging higher-order representations to faithfully express the geometry of the data, such as directionality in their intermediate layers. In this work, we propose an equivariant GNN that operates with Cartesian coordinates to incorporate directionality and we implement a novel attention mechanism, acting as a content and spatial dependent filter when propagating information between nodes. Our proposed message function processes vector features in a geometrically meaningful way by mixing existing vectors and creating new ones based on cross products. We demonstrate the efficacy of our architecture on accurately predicting properties of large biomolecules and show its computational advantage over recent methods which rely on irreducible representations by means of the spherical harmonics expansion.

## 1 Introduction

Predicting molecular properties is of central importance to applications in pharmaceutical research and protein design with the incentive to establish accurate computational methods to accelerate the overall process of finding better molecular candidates in a faster and cost-efficient way. Learning on 3D environments of molecular structures is a rapidly growing area of machine learning with promising applications but also domain-specific challenges. While Deep Learning (DL) has replaced hand-crafted features to a large extent, many advances are crucially determined through inductive biases in deep neural networks. Developed neural models should maintain an efficient and accurate representation of structures with even up to thousand of atoms and correctly reason about their 3D geometry independent of orientation and position. A powerful method to restrict a neural network to the functions of interest, such as a molecular property, is to exploit the *symmetry* of the data by constraining *equivariance* with respect to transformations from a certain symmetry group [1, 2].

3D Graph Neural Networks (GNNs) have been applied on a broad field involving molecular structures, such as in the prediction of quantum chemistry properties of small molecules [3, 4] and also on macromolecular structures like proteins [5–8] due to the natural representation of structures as graphs, with atoms as nodes and edges drawn based on bonding or spatial proximity. These networks

---

[*]Work was done during time at Bayer AG.

T. Le et al., Representation Learning on Biomolecular Structures using Equivariant Graph Attention. *Proceedings of the First Learning on Graphs Conference (LoG 2022)*, PMLR 198, Virtual Event, December 9–12, 2022.

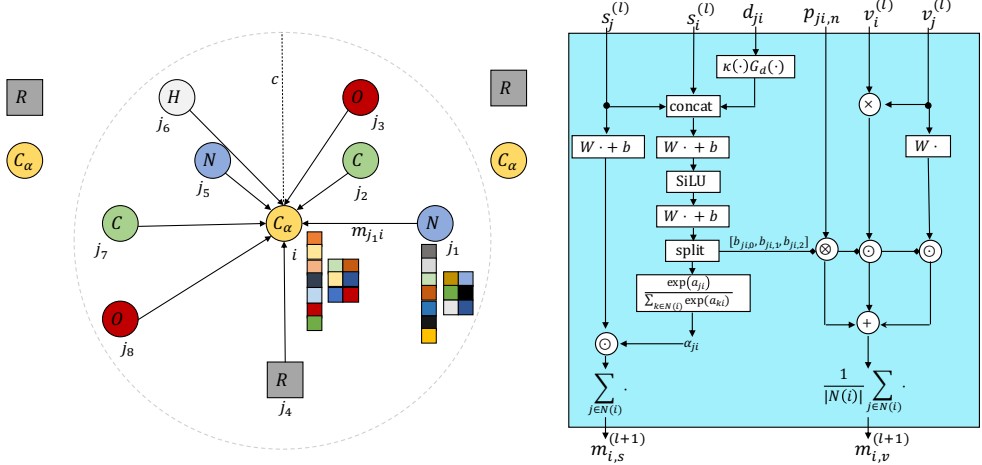

**(a)** Propagation flow for central node $i$.  **(b)** Proposed equivariant message function $M_l(\cdot)$.

**Figure 1:** (a) Visualization of the local neighbourhood of central carbon atom $i$. Directed edges illustrate the message flow from neighbour $j$ to central atom $i$, where scalar and vector features are propagated along the edges. Grey boxes $R$ represent the side-chain atoms of each residue and serve here as visual compression that include many more atoms. Here, nodes comprise scalar and vector features with 7 and 2 channels, respectively. (b) Proposed equivariant message function that computes a geometric and content related feature attention filter for scalar features, while vector messages are created based on a weighted combination of newly constructed vectors.

generally encode the 3D geometry in terms of rotationally invariant representations, such as pairwise distances to model local interactions which leads to a loss of directional information, while including angular information into network architecture has shown to be beneficial in representing the local geometry [9–11].

Neural models that preserve equivariance on point clouds in 3D space have been proposed [12–15] which can be described as Tensorfield Networks. These group-theoretic inspired models leverage higher-order representations by means of the spherical harmonics expansion of normalized relative positions to initially create equivariant features. While these models enable the interaction between different-order representations, (often referred to as type-$l$ representation), many data types are often restricted to scalar values (type-0 e.g., temperature or energy) and 3D vectors (type-1 e.g., velocity or forces). Another design choice is to define equivariant functions that directly operate on Cartesian coordinates [16–19], instead on the basis provided by the spherical harmonics. Following this approach, one could define (equivariant) transformations on Cartesian tensors, like rank 0 scalar(s) and rank 1 vector(s), which is the scope of this work and conceptually simpler and does not require Clebsch-Gordan tensor products of irreducible representations as commonly used in Tensorfield Network-like architectures.

In this work, we introduce Equivariant Graph Attention Networks (EQGAT) that operate on large point clouds such as proteins or protein-ligand complexes and show its superior performance compared to invariant models as well as our proposed model's faster training time compared to recent architectures that achieve equivariance through the usage of irreducible representations. Our model implements a novel feature attention mechanism which is invariant to global rotations and translations of inputs and includes spatial- but also content related information which serves as powerful edge embedding when propagating information in the Message Passing Neural Networks (MPNNs) [4] framework. Since we define equivariant functions on the original Cartesian space while restricting ourselves to tensor representations up to rank 1, i.e., scalars and vectors, we aim to capture as much geometrical information as possible through a geometrically motivated message function.

In summary, we make the following contributions:

- We introduce a computationally efficient equivariant Graph Neural Network that leverages geometric information by operating on vector features in Cartesian space.
- We implement a novel feature attention mechanism to propagate neighbouring node features and we define equivariant operations to combine vector features in a geometrically meaningful way.
- We benchmark our proposed architecture on large molecular systems such as protein complexes and show its efficacy mostly relevant to industrial applications.

## 2 Background

### 2.1 Message Passing Neural Networks (MPNNs)

MPNNs [4] generalize Graph Neural Networks (GNNs) [1, 2, 20] and aim to parameterize a mapping from a graph to a feature space. That feature space can either be defined on the node- or graph level. Formally, a graph $\mathcal{G} = (\mathcal{V}, \mathcal{E})$ contains nodes $i \in \mathcal{V}$ and edges $(j, i) \in \mathcal{E}$ which represent the relationship between nodes $j$ and $i$. Since MPNNs utilize shared trainable layers among nodes, permutation equivariance is preserved.

In this work, we consider graphs representing molecular systems embedded in 3D Euclidean space, where atoms represent nodes and the edges are described through covalent bonds and/or by atom pairs within a certain cutoff distance $c$ as illustrated in Figure 1(a). In the case of protein point clouds, a common design choice is the construction of residue graphs, where the nodes are represented through the $C_\alpha$-atom of each amino acid residue [5, 6, 18].

We refer $x_i^{(l)} = (a_i, p_i, s_i^{(l)}, v_i^{(l)})$ to the state of the $i-$th atom, where $a_i \in \mathbb{Z}_+$ and $p_i \in \mathbb{R}^3$ denote atom $i$'s chemical element and its spatial position, while $h_i^{(l)} = (s_i^{(l)}, v_i^{(l)}) \in \mathbb{R}^{1 \times F_s} \times \mathbb{R}^{3 \times F_v}$ are the hidden scalar and vector features that are iteratively refined through $L$ message passing steps. We distinguish between scalar and vector features because scalars can be transformed without functional restrictions, e.g., with standard MLPs, and their domain spans the entire $\mathbb{R}$, while vector features that reside in $\mathbb{R}^3$ can only be transformed in certain ways to preserve rotation equivariance. In theory, one could also only rely on vector features (with a number of $F_v$ channels), and perform a self-dot product reduction to make that representation invariant. This step however, restricts the domain space of scalars onto $\mathbb{R}_+$ only.

A general MPNN implements a learnable *message* and *update* function denoted as $M_l(\cdot)$ and $U_l(\cdot)$ to process atom $i-$th's hidden feature by considering its local environment $\mathcal{N}(i)$ through

$$m_i^{(l+1)} = \sum_{j \in \mathcal{N}(i)} M_l(x_i^{(l)}, x_j^{(l)}), \ \text{ and } \ x_i^{(l+1)} = (a_i, p_i, U_l(x_i^{(l)}, m_i^{(l+1)})), \tag{1}$$

where $\mathcal{N}(i) = \{j : ||p_{ij}||_2 = ||p_j - p_i||_2 = d_{ij} < c\}$ denotes central atom's $i-$th neighbour set that is obtained through a distance cutoff $c > 0$.

For our 3D GNN, we wish to implement simple, yet powerful rotation equivariant transformations in the message and update functions, to accurately describe the local environment of nodes in the point cloud.

### 2.2 Invariance and Equivariance

In this work, we consider the special orthogonal group SO(3), i.e. the group of proper rotations in three dimensions. A group element of SO(3) is commonly represented as matrix $R \in \mathbb{R}^{3 \times 3}$ satisfying $R^\top R = RR^\top = I$ and $\det R = 1$.
For a node feature $h = (s, v) \in \mathbb{R}^{F_s} \times \mathbb{R}^{3 \times F_v}$, an SO(3)-equivariant function $f(h) = h' = (s', v')$ must obey the following equation

$$f(g.h) = g.(s', v') = (Is', Rv') = (s', Rv') = g.f(h), \tag{2}$$

where $g.o$ in this work means, a group element $g$ of SO(3) acting on the object $o$. As shown in (2), invariance can be regarded as special case of equivariance, where equivariance for a scalar representation means that the *trivial* representation, i.e. the identity, acts on the scalar embedding, while vectors are transformed with $R$, i.e., a change of basis is performed, where the new basis is determined by the columns in $R$.

## 3    Related Work

Neural networks that specifically achieve E(3) or SE(3) equivariance have been proposed in Tensorfield Networks (TFNs) [12] and its variants in the covariant Cormorant [13], NequIP [15] and SE(3)-Transformer [14] which includes the attention mechanism in their architecture. With TFNs, equivariance is achieved through the usage of equivariant function spaces such as spherical harmonics combined with Clebsch-Gordan tensor products in their intermediate layer to allow the multiplication of different ordered representations, while others resort to lifting the spatial space to higher-dimensional spaces such as Lie group spaces [21]. Since no restriction on the order of representations is imposed on these methods, sufficient expressive power of these models is guaranteed, but at a cost of enlarged computational calculations with increased time and memory. It was recently analyzed by Brandstetter et al. [22] that the implementation of non-linear equivariant Graph Neural Networks in their model, which they term Steerable E(3) Equivariant Graph Neural Networks (SEGNN) achieves strong empirical results on small point clouds like the N-Body experiment or QM9 dataset, but also larger systems as in the OC20 dataset. One of their insights is that the construction of their (non-linear) SEGNN-layer, allows the model to better capture the local environment and enables the reduction of radius cutoff when constructing the neighbour list for each central atom $i$, since the Clebsch-Gordan tensor products between neighbouring nodes is computationally expensive. To circumvent the expensive computational cost, another line of research proposed to implement equivariant operations in the original Cartesian space, providing and efficient approach to preserve equivariance as introduced in the E($n$)-GNN [16], GVP [18, 23], PaiNN [17] and ET-Transformer [24] architectures without relying on irreducible representation of the orthogonal group by means of the spherical harmonics basis as originally introduced in TFN and implemented in the e3nn framework [25]. Aside of 3D atomistic GNNs, the attention mechanism has also been implemented in the GAT [26] and GATv2 [27] architectures, where GATv2 achieves superior performance over GAT due to the implementation of attention coefficients using a multilayer perceptron (MLP).

Our proposed model implements equivariant operations in the original Cartesian space and includes a continuous filter through the self-attention coefficients which serve as spatial- and content based edge embedding in the message propagation, as opposed to the PaiNN model where the filter solely depends on the distance. Additionally, our model constructs vector features from the given point cloud and leverages geometrical products that are efficient to compute. The E($n$)-GNN architecture does not learn vector features with several channels, but only updates a single vector feature[2] through a weighted linear combination, where the (learnable) scalar weights are obtained from invariant embeddings. The GVP model which was initially designed to work on macromolecular structures includes a complex message functions of concatenated node- and edge features composed with a series of GVP-blocks that enables information exchange between scalar and vector features, through dot product reduction of vectors, with a potential disadvantage of discontinuities through non-smooth components for distances close to the cutoff.

## 4    Proposed Model Architecture

### 4.1    Input Embedding

We initially embed atoms of small molecules or proteins based on their element/amino acid type using a trainable look-up table through $s_i^{(0)} = \text{embed}(a_i)$, which provides a starting (invariant) scalar representation of the node prior to the message passing. As in most cases, no initial vector features for atoms are available, we initialize them as zero tensor $v_i^{(0)} = 0 \in \mathbb{R}^{3 \times F_v}$.

### 4.2    Edge Filter through Feature Attention

For the two-body interaction between neighbouring node(s) $j$ to central node $i$, we implement a non-linear edge filter that depends on content related information stored in the scalar features $(s_j, s_i)$ and a radial basis expansion of the Euclidean distance $d_{ji} \leq c$. We choose the (orthonormal) Bessel basis $G_d : \mathbb{R} \to \mathbb{R}^K$ that projects the distance into $K$ basis values as introduced by Gasteiger et al. [9] and their polynomial envelope function $\kappa : [0, c] \to (0, 1]$ that smoothly transitions from 1 to 0 as

---

[2]In the E($n$)-GNN architecture, Cartesian coordinates of particles $p \in \mathbb{R}^3$ are updated.

the cutoff value $c$ is approached. The computation of the attention edge-filter is obtained through

$$e_{ji}^{(l+1)} = [s_i^{(l)}||s_j^{(l)}||\kappa(d_{ji})G_d(d_{ji})] \in \mathbb{R}^{2F_s+K}$$
$$f_{ji}^{(l+1)} = \text{MLP}(e_{ji}^{(l+1)}) \in \mathbb{R}^{F_s+3F_v}, \tag{3}$$

where MLP refers to an 1-layer Multilayer-Perceptron with SiLU activation function [28]. The input to the MLP is a concatenation of scalar features as well as a by $\kappa$ scaled radial basis expansion of the distance between nodes $j$ and $i$. The SO(3)-invariant embedding $f_{ji}^{(l+1)}$ represents the $F_s + 3F_v$ attention logits which are further split into $f_{ji}^{(l+1)} = [a_{ji}, b_{ji}]^{(l+1)}$ to be used as a non-linear filter when propagating neighbouring features. A novelty of our approach is that the attention coefficient between two vertices $j$ and $i$ is in fact obtained per feature-channel instead for the entire embedding as commonly achieved through a single scalar value, as done in GATv2 [27], albeit we also include edge-features through distances. The feature attention for the scalar embeddings is computed using the standard softmax activation function

$$\alpha_{ji} = \frac{\exp(a_{ji})}{\sum_{k \in \mathcal{N}(i)} \exp(a_{ki})} \in (0, 1)^{F_s}, \tag{4}$$

where the normalization in the denominator runs over all neighbours $k$ and the exponential function is applied componentwise. We choose to compute a non-linear intermediate edge-filter $f_{ji}$ due to increased expressivity through an 1-layer MLP. The embedding $b_{ji} \in \mathbb{R}^{3F_v}$ is processed to create coefficients that serve as weights for a linear combination of vector quantities to compute the vector message from $j$ to $i$, which we will describe in the following subsection.

## 4.3 Equivariant Message Propagation

We follow the idea of standard convolution, which is a linear transformation of the input, and compute the scalar features message for central node $i$ as

$$m_{i,s}^{(l+1)} = \sum_{j \in \mathcal{N}(i)} \alpha_{ji}^{(l+1)} \odot W_s^{(l+1)} s_j^{(l)}, \tag{5}$$

where $W_s^{(l+1)} \in \mathbb{R}^{F_s \times F_s}$ is a trainable weight matrix shared among all nodes and $\alpha_{ji}^{(l+1)}$ the non-linear attention filter obtained in (4).

In context of atomistic neural network potentials (NNPs), the filter $\alpha_{ji}^{(l+1)}$ is commonly implemented as an MLP that only inputs the distance $d_{ji}$ (by means of a radial basis expansion) as in SchNet [3], PaiNN [17], NequIP [15], while recent NNPs such as Allegro [29] and BOTNet [30] implement edge-filters that depend on the distance as well as node content, e.g., the chemical elements, unifying the idea of MPNNs in the context of machine learning force fields.

The recent work by Brandstetter et al. [22] analyzes modern 3D equivariant GNNs with the insight that non-linear message and non-linear update functions combined with their proposed *steerable* features space leads to an improved model, which they term SEGNN. The SEGNN, in similar spirit to Tensorfield Networks, can leverage higher-order equivariant representations up to a maximal rotation order $l_{\text{max}}$ through the spherical harmonics expansion of relative positions, which they take as steerable feature basis. Their proposed model implements *steerable* MLPs into the message- and update function to leverage non-linearity and geometric covariant information of the steerable features that go beyond $l = 0$, i.e., scalar features while our architecture is only restricted to scalar information, albeit vector information is still processed in the layers but then reduces to a scalar by a dot product operation. Our proposed message function for scalar features in Eq. (5) can also be formulated as a linear transformation where the weight matrix depends on distances but also hidden scalar information. To see this, we rewrite $\alpha_{ji}^{(l+1)} \in (0, 1)^{F_s}$ as matrix using the diagonal operator $A_{ji}^{(l+1)} = \text{diag}(\alpha_{ji}^{(l+1)}) \in (0, 1)^{F_s \times F_s}$ and observe that the filter scales the (independent) weight matrix $W_s^{(l+1)}$ leading to the message propagation

$$m_{i,s}^{(l+1)} = \sum_{j \in \mathcal{N}(i)} A_{ji}^{(l+1)} W_s^{(l+1)} s_j^{(l)} = \sum_{j \in \mathcal{N}(i)} W_{ji}^{(l+1)} s_j^{(l)},$$

where $W_{ji}^{(l+1)}$ defines the linear transformation matrix which depends on SO(3)-invariant information through $(s_i^{(l)}, s_j^{(l)}, d_{ji})$. The scalar message propagation can still be interpreted as non-linear convolution as the $A_{ji}^{(l+1)}$ weight matrix is obtained through an MLP and softmax activation function.

**Building Equivariant Features.** In many cases, no initial vector features are provided in raw point cloud data. However, when working with a protein backbone, i.e., the sequence of atoms $(C_\alpha, C, O, N)_i$, initial vectorial (node) features that describe the local environment of each backbone atom can be pre-computed as described by Ingraham et al. [6] and Jing et al. [18]. In a full-atom model, initial vector features for a node $i$ can be obtained by averaging over relative position vectors $v_{i,0} = \frac{1}{|\mathcal{N}(i)|} \sum_{j \in \mathcal{N}(i)} p_{ji} \in \mathbb{R}^3$ which satisfies Eq. (2) due to linearity. In our work, we initialize the vectors as zero tensor as described in Subsection 4.1 and calculate equivariant features by utilizing normalized relative positions $p_{ji,n}$ in the first layer to describe the directional interaction between central node $i$ and its neighbour $j$. In the subsequent layers, we extend the set of vectors by (1) constructing vectors based on normalized relative positions again, (2) mixing existing vector channels from the previous iteration, and (3) creating new vector quantities by making use of the cross product.

(1) We create equivariant vector features based on normalized relative position $p_{ji,n} = \frac{1}{d_{ji}}(p_i - p_j)$ as those provide directional information. Since we explicitly model scalar and vector features, each equipped with $F_s$ and $F_v$ channels, respectively, the tensor product offers a natural way to obtain a vector feature, by simply combining a vector and a scalar. Equivariant interactions between node $j$ and $i$ are computed through

$$v_{ji,0}^{(l+1)} = p_{ji,n} \otimes b_{ji,0}^{(l+1)} = p_{ji,n} b_{ji,0}^{(l+1)\top} \in \mathbb{R}^{3 \times F_v}, \tag{6}$$

which preserves SO(3) equivariance, due to the linearity of the tensor product. We note that the creation of 'initial' equivariant features in such manner is also performed in architectures, like [12, 13, 15, 22] just to name a few, that make use of irreducible representations of the SO(3) group by means of the spherical harmonics and implement the Clebsch-Gordan tensor product $(\otimes_{cg})$ that allows the mixing of possibly higher-order embedding representations of type $l > 1$, while we restrict ourselves to vector representations only, i.e. features of order $l = 1$ or equivalently Cartesian rank 1 tensors. The representation in Eq. (6) can be interpreted as $F_v$ scaled relative position vectors.

(2) In similar fashion to the (independent) linear transformation of scalar channels, we mix the vector channels using a learnable weight matrix $W_v^{(l)} \in \mathbb{R}^{F_v \times F_v}$ which preserves SO(3) equivariance due to the linearity property

$$v_n^{(l+1)} = v^{(l)} W_v^{(l+1)},$$

and is shared among all nodes. For a particular neighbouring node $j$, we scale the linearly transformed vectors

$$v_{ji,1}^{(l+1)} = b_{ji,1}^{(l+1)} \odot v_{n,j}^{(l+1)}, \tag{7}$$

which can be interpreted as a gating of previously mixed vectors.

(3) To capture more geometric information, while restricting the representation to be of rank 1, we utilize the vector cross product $c = (a \times b) \in \mathbb{R}^3$ between two vectors $a$ and $b$ that satisfy the following rotation invariance property

$$Ra \times Rb = R(a \times b).$$

The output of the cross product $a \times b$ defines a vector $c$ that is perpendicular to plane spanned by $a$ and $b$. Here, we calculate the cross product on the same channels from the previous layer vector features of node $i$ and $j$ as

$$\tilde{v}_{ji,2}^{(l+1)} = (v_i^{(l)} \times v_j^{(l)}) \in \mathbb{R}^{3 \times F_v},$$

to reduce the computational complexity.

We highlight that recent equivariant GNNs which work with rank 1 Cartesian tensors, such as GVP, PaiNN or ET-Transformer do not include the cross product in their architecture and are restricted in the creation of vector features that may span the entire $\mathbb{R}^3$. These architecture make use of step (1) and (2) only. For example, when all atoms are placed on the $xy$-plane, using step (1) and (2) would

always create vectors on the $xy$ plane, while the coordinate on $z$ axis is always $0$. By leveraging the cross product, vectors in the $z$ direction can be computed, without increasing the rank order[3].

We note that our assumption on SO(3) equivariance is attributed to the fact of using the cross product in our architecture. For the case that practitioners care about O(3) equivariance, our proposed EQGAT might be suboptimal for usage since we do not distinguish polar or pseudo vectors in the internal network representation. If O(3) equivariance is desired, special care on the selection between input vectors in the cross product have to be made, in order to correctly assign the output parity type. E.g., a cross product of two polar vectors will return a pseudo vector, while a cross product of a polar and pseudo vector will return a polar vector.

In similar fashion to Eq. (6) and (7), each channel of the representation $\tilde{v}_{ji,2}^{(l)}$ is weighted by the SO(3) non-linear filter $b_{ji,2}^{(l)} \in \mathbb{R}^{F_v}$ to obtain

$$v_{ji,2}^{(l+1)} = b_{ji,2}^{(l+1)} \odot \tilde{v}_{ji,2}^{(l+1)}, \tag{8}$$

Finally, we define the vector message from node $j$ to central node $i$ as the sum of the three components in (6) to (8) and aggregate it across all neighbouring nodes $j \in \mathcal{N}(i)$ to obtain the vector message

$$m_{i,v}^{(l+1)} = \frac{1}{|\mathcal{N}(i)|} \sum_{j \in \mathcal{N}(i)} (v_{ji,0}^{(l+1)} + v_{ji,1}^{(l+1)} + v_{ji,2}^{(l+1)}), \tag{9}$$

which results into new weighted geometric vectors by utilizing the (static) relative positions as well as neighbouring vector features and lastly, normal vectors obtained through the cross product. Since we combine the three vector components through a gating mechanism, we do not use an attention mechanism on vector features to avoid additional computational steps and the fact that the calculation of attention logits had to be done using some SO(3) invariant input, which would make the model more complicated. We provide the full proof of SO(3) equivariance of Eq. (9) in Appendix C.

**Equivariant Update Function.** After obtaining the aggregated message for central node $i$ in the representation $m^{(l+1)} \in \mathbb{R}^{F_s} \times \mathbb{R}^{3 \times F_v}$, we implement a residual connection as intermediate update step

$$\tilde{s}_i^{(l+1)} = s_i^{(l)} + m_{i,s}^{(l+1)}, \text{ and } \tilde{v}_i^{(l+1)} = v_i^{(l)} + m_{i,v}^{(l+1)}$$

while in the update layer, we implement an equivariant non-linear transformation inspired by gated non-linearities proposed by [31] and used in [17] with minor modification as shown in Figure 2. Notably, the scalar features receive geometric information by concatenating the norm of linear transformed vector features, while the 1-layer scalar MLP is tasked to transform the combined embeddings to update the scalar states and retrieve non-linear weights that are used to reweight vector features. We apply these weights by element-wise multiplying with linearly transformed vector features as shown on the right which can also be interpreted as variants of the Gated Linear Unit [32, 33], followed by a linear layer to implement an equivariant MLP for vector features.

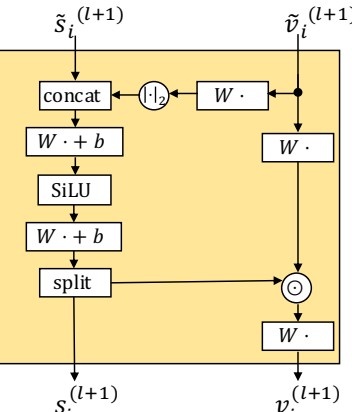

**Figure 2:** A gated equivariant MLP that transforms scalar and vector features into a new representation. Here we used this block as update function $U_l(\cdot)$.

## 5 Experiments and Results

We test the efficacy of our proposed EQGAT model on five publicly available molecular benchmark datasets which pose significant challenges for the development of efficient and accurate prediction models in protein design.

---

[3]Two rank 1 Cartesian tensors, i.e., two vectors can also be combined by computing the tensor product of the two, which results into a rank 2 Cartesian tensor with 9 elements in the matrix. This rank 2 Cartesian tensor contains 3 unique elements of the cross product in its antisymmetric part after a sum decomposition.

**Table 1:** Benchmark results on ATOM3D tasks. We report the results for the Baseline models from [34] and GVP-GNN [23]. We run our own experiments with the SchNet, PaiNN, SEGNN and our EQGAT model and report averaged metrics over 3 runs. For the SEGNN model we only report the results on a single run due longer training time. $R_S$ stands for Spearman Rank Correlation, RMSE abbreviates Root Mean Square Deviation and ROCAUC the area under ROC curve.

| Tasks | PSR (↑) | | RSR (↑) | | LBA (↓) | RES (↑) | PPI (↑) |
|---|---|---|---|---|---|---|---|
| Metric | Mean $R_S$ | Global $R_S$ | Mean $R_S$ | Global $R_S$ | RMSE | Accuracy | ROCAUC |
| CNN | $0.431 \pm 0.013$ | $0.789 \pm 0.017$ | $0.264 \pm 0.046$ | $0.372 \pm 0.027$ | $\mathbf{1.416 \pm 0.021}$ | $0.451 \pm 0.002$ | $0.844 \pm 0.002$ |
| GNN | $\mathbf{0.515 \pm 0.010}$ | $0.755 \pm 0.004$ | $0.234 \pm 0.006$ | $\mathbf{0.512 \pm 0.049}$ | $1.570 \pm 0.025$ | $0.082 \pm 0.002$ | $0.669 \pm 0.001$ |
| GVP-GNN | $0.511 \pm 0.010$ | $0.845 \pm 0.008$ | $0.211 \pm 0.142$ | $0.330 \pm 0.054$ | $1.594 \pm 0.073$ | $0.527 \pm 0.003$ | $0.866 \pm 0.004$ |
| SchNet | $0.448 \pm 0.016$ | $0.784 \pm 0.013$ | $0.247 \pm 0.029$ | $0.273 \pm 0.017$ | $1.522 \pm 0.015$ | $0.326 \pm 0.003$ | $0.839 \pm 0.005$ |
| PaiNN | $0.462 \pm 0.015$ | $0.809 \pm 0.003$ | $0.270 \pm 0.062$ | $0.462 \pm 0.064$ | $1.507 \pm 0.033$ | $0.370 \pm 0.004$ | $0.884 \pm 0.002$ |
| SEGNN | $0.474$ | $0.833$ | $-0.099$ | $0.252$ | $1.450 \pm 0.011$ | $0.454$ | $0.854$ |
| EQGAT | $0.491 \pm 0.008$ | $\mathbf{0.847 \pm 0.006}$ | $\mathbf{0.316 \pm 0.029}$ | $0.404 \pm 0.096$ | $1.440 \pm 0.027$ | $\mathbf{0.540 \pm 0.017}$ | $\mathbf{0.908 \pm 0.001}$ |

## 5.1 ATOM3D

The ATOM3D benchmark [34] provides datasets for representation learning on atomic-level 3D molecular structures of different kinds, i.e., proteins, RNAs, small molecules and complexes. Since proteins perform specific biological functions essential for all living organisms and hence, play a key role when investigating the most fundamental questions in the life sciences, we focus our experiments on the learning problems often encountered in structural biology with different difficulties due to data scarcity and varying structural sizes. We use provided training, validation and test splits from ATOM3D and refer the interested reader to the original work of Townshend et al. [34] for more details. For all benchmarks, we compare against the Baseline CNN and GNN models provided by Townshend et al. [34] from ATOM3D, GVP-GNN reported in [23] and we run experiments for SchNet [3], an SO(3) invariant GNN architecture that has shown strong performance on small molecule prediction tasks, PaiNN [17] as SchNet's improved SO(3) equivariant architecture and the recently proposed SEGNN [22] that leverages higher-order representations by means of the irreducible representations and Clebsch-Gordan tensor products using their official code base.

For SchNet, PaiNN and our proposed EQGAT architecture, we implement a 5-layer GNN with $F_s = 100$ scalar channels and $F_v = 16$ vector channels for the PSR, RSR, RES and PPI benchmark, as these benchmarks consists of more training samples and comprise larger biomolecules. For the Ligand Binding Affinity (LBA) task, we utilize a 3-layer GNN with the same number of scalar- and vector channels. For the SEGNN architecture, we implement a 3-layer GNN with $(100, 16, 8)$ channels for the embeddings of type $l = (0, 1, 2)$ that transform according to the irreducible representation of that order preserving SO(3) equivariance. The edges in the point clouds are constructed based on a radius cutoff of $4.5\text{Å}$. All graphs are considered as full-atom graphs, i.e., the initial node feature is determined by the chemical element.

The Protein and RNA Structure Ranking tasks (PSR / RSR) in ATOM3D are both regression tasks with the objective to predict the quality score in terms of *Global Distance Test* (GDT_TS) or Root-Mean-Square Deviation (RMSD) for generated Protein and RNA models wrt. to its experimentally determined ground-truth structure. The ability to reliably rank a biopolymer structure requires a model to accurately learn the atomic environments such that discrepancies between a ground truth states an its corrupted version can be distinguished. We evaluated our model on the biopolymer ranking and obtained good results on the current benchmark, as reported in Table 1 in terms of Spearman rank correlation. Our proposed model performs particularly well on the PSR task outperforming the GVP-GNN [23] on the Global Rank Spearman correlation on the test set, while our model is more parameter efficient (383K vs. 640K). We believe our model could be further improved by additional hyperparameter tuning, e.g., by increasing the number of scalar or vector channels, which we did not do in our study to compare against the baseline models.

We noticed that the RSR benchmark was particularly difficult to validate as only a few dozen experimentally determined RNA structures are existent to date, and the structural models generated in the ATOM3D framework are labeled with the RMSD to its native structure, which is known to be sensitive to outlier regions, for exampling by inadequate modelling of loop regions [35], while the GDT_TS metric might be a better suited target to predict a ranking for generated RNA structures as in the PSR benchmark.

Another challenging and important task for drug discovery projects is estimating the binding strength (affinity) of a candidate drug atomistic's interaction with a target protein. We use the ligand binding affinity (LBA) dataset and found that among the GNN architectures, our proposed model obtains the best results, while also being computationally cheap and fast to train. The best performing model in the LBA-task is a 3D CNN model which works on the joint protein-ligand representation using voxel space and enforcing equivariance through data augmentation. The inferior performance of all equivariant GNNs might be caused by the need of larger filters to better capture the locality and many-body effects, where 3D CNNs have an advantage when using voxel representations, while GNNs commonly capture 2-body effects. Furthermore, as all GNN models jointly represent ligand- and protein as *one* graph by connecting vertices through a distance cutoff of $4.5\text{Å}$, we believe that such union leads to an information loss of distinguishing the atom identity from the ligand and protein. A promising direction to investigate is to incorporate a ligand and protein GNN encoder seperately and merge the two embeddings prior the binding affinity prediction, similar to Graph Matching Networks [36] and recently realized by Stärk et al. [37] in a slightly different context.

EQGAT outperforms the current SOTA GVP-GNN model on the *Residue* and *Protein-Protein-Interaction* benchmarks which are both node classification tasks and require a model to accurately capture the local environment of a selected $C_\alpha$ atom to serve as expressive input for a downstream (decoder) network to obtain the final prediction.

Notably, our proposed EQGAT architecture performs on par with the SEGNN that implements internal representations of higher order, i.e., of rotation order up to $l = 2$. We believe that including the cross product in our vector message in (9) allows the model to capture more geometric detail in a possible protein ligand binding pose for accurately predicting the binding affinity, which is investigated in the following ablations.

## 5.2  Ablation Studies

To evaluate the benefits of our designed EQGAT architecture, we perform ablation studies and remove architectural components to isolate the effect of each design choice on performance.

**Table 2:** Results of the ablation studies.

|  | LBA [RMSE ↓] | PSR [Mean \| Global $R_S$ ↑] |
|---|---|---|
| No-Cross-Product | 1.458 (0.011) | 0.477 (0.012) \| 0.827 (0.010) |
| No-Feature-Attention | 1.466 (0.040) | **0.492 (0.007)** \| 0.820 (0.002) |
| Full Model | **1.440 (0.027)** | 0.491 (0.008) \| **0.847 (0.006)** |

Ablation study 1 (termed No-Cross-Product) removes the contribution of vector cross product (denoted as $v_{ji,2}$ in Eq. (9)). This leads to the effect that the vector message is solely constructed based on scaled versions of normalized relative positions ($v_{ji,0}$) and linear combinations of existing vector features ($v_{ji,1}$).

Ablation study 2 (termed No-Feature-Attention) replaces the feature attention coefficient $\alpha_{ji} \in (0,1)^{F_s}$ through a single coefficient $\alpha_{ji} \in (0,1)$.

We observe that the full EQGAT architecture obtains the best performance among the two datasets compared to the ablated models although we note that the improved performance of the full model in RMSE on the LBA benchmark and Global $R_S$ in the PSR benchmark is difficult to attribute to the inclusion of architectural components due to the (larger) variance obtained through the 3 runs for each experiment.

## 6  Conclusion, Limitations and Future Work

In this work, we introduced a novel attention-based equivariant graph neural network for the prediction of properties of large biomolecules that achieves superior performance on the ATOM3D benchmark. Our proposed architecture makes use of rotationally equivariant features in their intermediate layers to faithfully represent the geometry of the data, while being computationally efficient, as all equivariant functions are directly implemented in the original Cartesian space without changing the representation through the spherical harmonics basis as commonly done in Tensorfield networks. As our proposed model operates on Cartesian tensors and we restrict the representation to be of rank 1 only, a general promising future direction of investigation is the implementation of Cartesian equivariant GNNs that

leverage higher-rank tensors in their layers, that are specifically implemented for learning purposes involving large biomolecules. As it is up to date not clear, how much improvement higher-order Cartesian tensors benefit for learning tasks that involve large biomolecular systems, we hope that our work and open-source code will be useful for the graph learning and computational biology community.

## Code Availability

We provide the implementation of our model and experiments on `https://github.com/Bayer-Group/eqgat`. We use PyTorch [38] as Deep Learning framework and PyTorch Geometric [39] to implement our GNNs.

## Author Contributions

T.L designed the model architecture, carried out the implementation and executed all experiments. T.L wrote the manuscript with input from F.N and D.A.C who both proofread the final manuscript. D.A.C supervised the work.

## Acknowledgements

The authors are grateful to the anonymous reviewers of the LoG conference 2022 for proof-reading the paper and suggesting improvements. This work was supported by several research grants. F.N acknowledges funding from the European Commission (ERC CoG 772230 'ScaleCell'), MATH+ (AA1-6), Math AA1-10 and Deutsche Forschungsgemeinschaft (CRC1114/B08). D.A.C acknowledged funding from the Bayer AG Life Science Collaboration ('DeepMinDS'). D.A.C also received financial support from European Commission grant numbers 963845 and 956832 under the Horizon2020 Framework Program for Research and Innovation. T.L acknowledges funding from Bayer AG's PhD scholarship.

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

## A   Appendix

## Full Model Details and Hyperparameters

All EQGAT models in this paper were trained on a single Nvidia Tesla V100 GPU.

**Table 3:** Description of architectural parameters on the ATOM3D benchmarks.

| Parameter | LBA | PSR | RSR |
|---|---|---|---|
| Learning rate (lr.) | $10^{-4}$ | $10^{-4}$ | $10^{-4}$ |
| Maximum epochs | 20 | 30 | 30 |
| Lr. patience | 10 | 10 | 10 |
| Lr. decay factor | 0.75 | 0.75 | 0.75 |
| Batch size | 16 | 16 | 16 |
| Num. layers | 3 | 5 | 5 |
| Num. RBFs | 32 | 32 | 32 |
| Cutoff [Å] | 4.5 | 4.5 | 4.5 |
| Scalar channels $F_s$ | 100 | 100 | 100 |
| Vector channels $F_v$ | 16 | 16 | 16 |
| Num. parameters | 238k | 383k | 383k |

We used the ADAM optimizer [40] apart from the defined learning rate all other standard hyperparameter setting from the PyTorch library.

## B   Model Efficiency

**Model Efficiency.**   We assess the model efficiency of EQGAT in terms of computation time as well as trainable parameters and compare against SchNet, PaiNN and SEGNN on the LBA, PSR and RSR benchmarks. These datasets have on average 408, 1624, and 2390 nodes per graph with 9180, 26756 and 44233 directed edges, respectively for the training set of LBA, PSR and RSR.

As these datasets consist of graphs with up to thousands of atoms, computationally- and memory efficient models are preferred such that batches of graphs can be stored on GPU memory and processed fast during training. We measure the inference time of a random batch comprising 10 macromolecular structures on an NVIDIA V100 GPU. As shown in Table 4, SchNet and

**Table 4:** Comparison on model efficiency when passing a batch of 10 macromolecular structures.

| Dataset | Model (# Param.) | Inference Time [ms] |
|---|---|---|
| LBA | EQGAT (238K) | 11.94 |
|  | SchNet (240K) | 8.25 |
|  | PaiNN (379K) | 10.66 |
|  | SEGNN (238K) | 89.53 |
| PSR | EQGAT (383K) | 49.96 |
|  | SchNet (240K) | 18.36 |
|  | PaiNN (379K) | 18.58 |
|  | SEGNN (238K) | 255.44 |
| RSR | EQGAT (383K) | 75.45 |
|  | SchNet (240K) | 27.27 |
|  | PaiNN (379K) | 26.98 |
|  | SEGNN (238K) | 390.69 |

PaiNN are both parameter efficient and both achieve the fastest inference time on a forward pass, while our proposed EQGAT is slower mainly due to the softmax attention normalization in the denominator in Eq. (4) which could be improved when the softmax attention with its normalization is replaced by a sigmoid activation function, to obtain soft-attention weights. This step however, results into a edge-filter $\alpha_{ji}$ that does not sum up to 1 when iterating over all neighbours $j$. The SEGNN model has the longest runtime on the forward pass across the 3 datasets. This is mostly attributed to the Clebsch-Cordan tensor products which can be very expensive in learning tasks that involve proteins, as the CG product is always performed on edges.

## C   Proof Equivariance

We prove the rotation equivariance in Eq. (9) which consists of the sum of three vector components, and displayed here again

$$m_{i,v}^{(l+1)} = \frac{1}{|\mathcal{N}(i)|} \sum_{j \in \mathcal{N}(i)} (v_{ji,0}^{(l+1)} + v_{ji,1}^{(l+1)} + v_{ji,2}^{(l+1)}).$$

As the sum is a linear function, we require to show that each summand $(v_{ji,0}, v_{ji,1}, v_{ji,2})$ is equivariant. For brevity, we omit all top indices. The first term is computed as tensor product of an $l = 1$ representation and $l = 0$ representation through

$$v_{ji,0} = p_{ji,n} \otimes b_{ji,0} = p_{ji,n} b_{ji,0}^{\top} \in \mathbb{R}^{3 \times F_v},$$

where $b_{ji,0} \in \mathbb{R}^{F_v}$ is an SO(3)-invariant representation, i.e. a scalar representation with $F_v$ channels, and $p_{ji,n} \in S_2 \subset \mathbb{R}^3$ a normalized relative vector, which lies on the 2-dimensional sphere. If the point cloud is rotated, as defined in Eq. (2), (relative) position as well as vector features change to

$$p \xrightarrow{R} Rp \,,$$

$$v \xrightarrow{R} Rv \,,$$

while the cross product between two vector features $v_0, v_1$ is invariant to rotation, resulting to the property

$$(Rv_0 \times Rv_1) = R(v_0 \times v_1) \,.$$

In case a rotation is acting on the system, from Eq. (2) we know how vector and scalar quantities transform, resulting into:

$$R.v_{ji,0} \to Rp_{ji,n} \otimes b_{ji,0} = R(p_{ji,n} \otimes b_{ji,0}) = Rv_{ji,0}.$$

due to the linearity of the tensor product which proves SO(3) equivariance for the first term. For the second term, we calculate

$$v_{ji,1} = b_{ji,1} \odot (v_i \times v_j),$$

where $b_{ji,1} \in \mathbb{R}^{F_v}$ is an SO(3)-invariant representation and the output of the cross product is a vector representation $\in \mathbb{R}^{3 \times F_v}$. To be precise, the elementwise multiplication from the left with the $b_{ji,1}$ has to be rewritten, to match the shape, i.e. unsqueeze a new dimension to scale each of the $F_v$ vector by the scalar value, resulting into:

$$v_{ji,1} = (1 \otimes b_{ji,1}) \odot (v_i \times v_j),$$

where 1 is the one-vector in 3 dimensions. For a rotation acting on the system, we conclude that

$$\begin{aligned} R.v_{ji,1} &\to (1 \otimes b_{ji,1}) \odot (Rv_i \times Rv_j) \\ &= (1 \otimes b_{ji,1}) \odot R(v_i \times v_j) = R(1 \otimes b_{ji,1}) \odot (v_i \times v_j) \\ &= Rv_{ji,1}, \end{aligned}$$

which proves SO(3) equivariance for the second term. The third term is obtained through

$$v_{ji,2} = (1 \otimes b_{ji,2}) \odot (v_j W_n),$$

where $b_{ji,2} \in \mathbb{R}^{F_v}$ is a scalar representation with $F_v$ channels and $W_n$ a linear transformation of shape $(F_v \times F_v)$. Due to linearity, we can see that

$$Rv_j W_n = (Rv_j) W_n = R(v_j W_n)$$

is SO(3) equivariant. As we elementwise multiply with a unsqueezed/expanded scalar representation, we conclude for the last term SO(3) equivariance

$$\begin{aligned} R.v_{ji,2} &\to (1 \otimes b_{ji,1}) \odot (Rv_j) W_n \\ &= (1 \otimes b_{ji,1}) \odot R(v_j W_n) = R(1 \otimes b_{ji,1}) \odot (v_j W_n) \\ &= Rv_{ji,2}. \end{aligned}$$

Since all three components in the sum are SO(3) equivariant, we conclude that the final sum is also SO(3) equivariant.

As the reader might have noticed, we build equivariant features based on linear functions and weighting $l = 1$ representations through $l = 0$ representations. This typical scaling is achieved through the tensor product $\otimes$. Our architecure however, also performs a multiplication between two $l = 1$ representations, through the cross product, which has the pleasant SO(3) invariance property that we can exploit to prove SO(3) equivariance, when scaling the output with an $l = 0$ representation.

**A Note on Translation Equivariance.** Our proposed model is translation invariant, as all vector features are initially created by means of a tensor product of (normalized) relative position $p_{ji,n}$. To see that, for any translation vector $t \in \mathbb{R}^3$ for relative positions, we can see that the calculation of such vectors[4] $p_{ji} = p_j - p_i$, are inherently translation invariant due to

$$t.p_{ji} \rightarrow (p_j + t) - (p_i + t) = p_j - p_i + t - t = p_j - p_i = p_{ji}.$$

Since we do not model absolute Cartesian coordinates, e.g., by updating the spatial coordinates through our layers, our model is not SE(3)-equivariant, i.e. next to rotation equivariance, also translation equivariant. We note that translation equivariance, however can be achieved through a simple operation such as the addition of an SE(3) representation with an SO(3) representation, e.g.

$$p_i = p_i + p_{ji,n} \otimes s,$$

where $s \in \mathbb{R}$ and reminiscent in the E($n$)-GNN architecture, albeit the authors are not using the notation of the tensor product.

## D  Synthetic Dataset

We adopt the synthetic dataset from GVP [18] with slight modifications to make it a more challenging task. We create 50,000 'structures' where each 'structure' consists of $n = 100$ random points in $\mathbb{R}^3$, distributed uniformly in the ball of radius $r = 10$ with the constraint that no two points are less than distance $d = 2$ apart. Three points are randomly chosen and are labelled as 'special' which will define the vertices of a triangle. The learning task is a multitask regression of 3 targets, where the first target is to predict the distance between the center of mass (COM) of the entire structure and the COM of the triangles spanned by the three special points. The second and third task is the prediction of the perimeter and surface area of the triangle. The choice of the 3 targets refers to a structural learning task, where the model requires to learn about the global shape of the structure, while the second and third targets are relational. An example structure is depicted in Figure 3. The evaluation metric is the MSE of the three tasks. We split the dataset into 80% training, 10% validation and 10% test sets.

**Table 5:** Evaluation of our proposed EQGAT architecture on Triangle benchmark.

| Model | Triangle [MSE ↓] | No. Params [$10^3$] |
|---|---|---|
| SchNet | 37.545 (1.838) | 16.8 |
| PaiNN | 10.259 (0.949) | 27.1 |
| SEGNN | **3.875 (0.879)** | 60.9 |
| GVP | 10.115 (1.210) | 61.6 |
| EQGAT-Full | 6.003 (0.432) | 27.4 |
| EQGAT-No-Cross-Product | 6.835 (1.066) | 27.4 |
| EQGAT-No-Feature-Attention | 6.808 (0.326) | 27.4 |

For the synthetic task of multitask regression we notice that the SEGNN architecture equipped with higher-order equivariant features up to rotation order 2, obtains the best performance, followed by our proposed EQGAT model that only incorporates rank 1 (vector) features. For the synthetic dataset, we did not perform any hyperparameter tuning and set the number of layers to 3 with $F_s = 32$ scalar

---

[4]We omit the normalization to unit vectors for brevity.

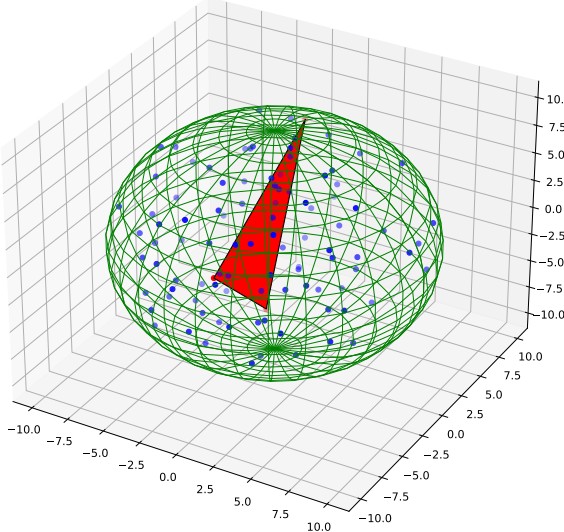

**Figure 3:** An example structure of the synthetic dataset. Three random points in the structure determine the vertices of a triangle, which is colored in red.

and $F_v = 8$ vector channels and train for 50 epochs. The number of trainable parameters for SchNet, PaiNN, SEGNN and EQGAT on the synthetic Triangle dataset are listed in the last column of Table 5.

