# OpenReview forum: "Representation Learning on Biomolecular Structures using Equivariant Graph Attention"
_logconference.io/LOG/2022/Conference — LoG 2022 Poster_

### Official Review · Reviewer_CRBR · 2022-10-18

**Overall Score:** 6
**Confidence:** 4

**Review:**

Summary
This work introduces a SE(3)-Equivariant MPNN for the prediction of properties of large biomolecules. A feature attention mechanism was proposed to propagate neighboring node scaler features. For vector features, it proposes a cross-product operation that implements in rank 1 Cartesian tensors and maintains the computational efficiency. The experimental results partially verify the value of the proposed modules.

Positives: 1. The paper is well-written and easy to follow. 2. This work summarizes the previous work in message passing, points out their disadvantages in vector features and the computational cost and deals with some of the problems.

Negatives:
* The feature attention is somewhat incremental. The authors should explain more about the motivations for using the scalar feature attention.
* Another major concern is the lack of experimental results. There are no ablation experiments in the paper, and there is no in-depth discussion and analysis of Atom3D subtask. Not that the results of EQGAT are not very consistent (only tops in 2/5 picked metrics), and the improvement is not significant (on average).


Questions：
* Please add some ablation study and analysis to explicitly show and verify your motivations.
* The reviewer is confused about the higher-rank Cartesian tensors. Since SEGNN uses the rank 2 tensors while the performance of EQGAT is better than SEGNN in all benchmarks. In this sense, how could you say the higher rank tensor can improve the network, and what is the difference in Cartesian tensors operation between this work and SEGNN?

Response to the authors:
I have read the responses and other reviews. The authors have resolved most of my raised issues and I appreciated the efforts in the additional experiments and explanations on cartesian tensors. Hence, I vote for a weak acceptance and will update my score from 3 to 6.

---

### Official Review · Reviewer_359T · 2022-10-21

**Overall Score:** 5
**Confidence:** 4

**Review:**

### Summary

The paper proposes a graph attention / message-passing mechanism (EQGAT) for learning SE(3)-invariant/equivariant scalar and vector features from graphs in 3D space, such as biomolecular structures. The method uses scalar node and edge features to compute attention weights over scalar messages, and forms vector messages via 3 mechanisms: (1) mixing in the edge vector itself, (2) mixing the node vector features, and (3) mixing the cross product of source and destination node vector features. The method is benchmarked on ATOM3D, where it does not meaningfully outperform the baselines.

### Pros

* The work ventures into the design space of "lightweight" equivariant graph neural networks, which have received less attention than their tensor-product based counterparts.

* EQGAT is the first lightweight equivariant GNN that incorporates the cross product in its message passing block, which expands the expressive power beyond previous works like EGNN and GVP. To my knowledge, it is also the first to incorporate graph attention.

### Cons

* The technical novelty and contributions of the work are quite limited relative to previous and well-established works (EGNN/GVP/PaiNN/ET-Transformer); largely amounting to a construction of an alternative (but not particularly novel) message passing block using simple and well-understood operations.

* The authors do not sufficiently motivate or discuss the implications of the cross-product operation. Having a cross-product has a very precise meaning in terms of giving up parity equivariance, which is a specific design choice that should be discussed. Previous lightweight GNNs satisfied parity equivariance, whereas tensor-product networks such as e3nn provide a clean way to separate invariant from equivariant outputs. EQGAT sweeps this issue under the rug and provides neither.

* The experiments are limited and fail to show any meaningful improvement of the proposed EQGAT over the baseline GNN. The runtime of the method is also noticeably slower than PaiNN, which blunts the paper's efficiency claims.

* There are areas of poor writing quality and inaccuracies in the paper which impair its clarity and message.

### Recommendation

I recommend rejection of the paper because the field of equivariant GNNs is populated and mature enough that introducing an alternative message-passing block, without particularly novel operations or insights or strong empirical results, does not constitute a significant technical contribution. The work represents only a minor advance over the 3 to 4 most similar well-established works in the field and is below the bar for a full-length submission.

### Presentation and Feedback

* The distinction between tensor-product based networks and those that "explicitly define the (equivariant) operations" is overstressed, as the latter are (aside from the information flow) computationally equivalent to tensor products with maximum rotation order 1. It may be more correct (abeit subjective) to merely claim that the former are harder to _understand_.

* The implication that EQGAT operates in "Cartesian space" while tensor-product methods do not is generally a misleading distinction. The emphasis on "content awareness" is also misleading, as it is standard to consider the node features in message-passing.

* There are several paragraph breaks that are formatted incorrectly (73, 86, 91, 158)

* Typos: 57 "capture as much"; 65 "show its efficacy"

* It is not true that dot products are restricted to positive numbers (85); this is true of _self_ dot-products only.

* The background on the very basics of MPNNs could be significantly abridged or even omitted for this audience.

* Equation 3 seems wrong: it should state $f(s, v) = s', v'$ implies $f(Is, Rv) = Is', Rv'$.

* Paragraph 172-193 is too long, and furthermore I do not see the point of emphasizing that EQGAT is akin to pseudo-linear transformation.

* I recommend removing Figure 1a as it doesn't seem to explain anything more than the basic concept of message passing. I would instead make Figure 1b larger and label more of the intermediate variables used in the main text.

---

### Official Review · Reviewer_Chvz · 2022-10-21

**Overall Score:** 6
**Confidence:** 3

**Review:**

The authors present EQGAT, an equivariant GNN architecture that operates on biological structures embedded in Euclidean Space. The model is evaluated on a collection of tasks from Atom3D and demonstrates reasonable performance with the authors highlighting inference time improvements relative to higher-order tensorfield network models. The proposed model has the capability of working with multiple channels of vector features, distinguishing it from the previous work on E(n)-GNNs by Satorras et al.

Firstly, the writing of the paper is excellent; it was really quite enjoyable to read. I commend the authors for providing clear motivation and synthesis of related work in such a clear manner.

## Suggestions
The selection of tasks for evaluation puzzles this reviewer a little. These tasks have a strong motivation for invariance, rather than equivariance (I understand invariant functions are a subset of equivariant functions) being the desired property with the exception of possibly the LBA task (but not under the protein-ligand graph formulation the authors have presented, and, strictly one probably desires non-equivariance here). Perhaps I am missing something, but I think the authors would make a stronger case evaluating the proposed method on a task where equivariance is of greater importance e.g. protein-ligand or protein-protein docking and invariant methods should fail by comparison.

Additionally, the protein/biomolecular structure representation learning space is beginning to mature with a plethora of protein-specific encoders being developed. This reviewer wonders why these were not examined in greater depth, compared to trivial baselines such as a CNN.

Overall, the evaluation feels lighter than this reviewer would like to see. However, the method is well-motivated and appears to work on par with existing methods so I’d recommend this work for acceptance.

### Minor
L196. The backbone is usually defined as N, C$\alpha$, C, O (not N, C$\alpha$, C$\beta$, O as the authors state). C$\beta$ is the first atom of a sidechain (eg GLY has no Cb). Could the authors provide some insight here?


## Update
I thank the authors for their thorough responses and the inclusion of additional experiments. I would raise my score to a 7 if the option to do so was available.

---

### Official Review · Reviewer_aj5D · 2022-10-25

**Overall Score:** 6
**Confidence:** 4

**Review:**

**Summary**:
The authors propose a $SO(3)$-equivariant message passing graph neural network model, termed *EQGAT*. *EQGAT* operates on scalars and Cartesian vectors. It employs an additive attention mechanism on scalar and distance features in its message passing scheme. For its equivariant vector message construction, it uses linear combinations of (1) Cartesian vector features, (2) normalized relative position vectors and (3) cross products between Cartesian vector features.
The model is evaluated on three tasks of the atom3d benchmark: Protein structure ranking (PSR), RNA structure ranking (RSR) and protein-ligand binding affinity prediction (LBA). On the studied benchmarks, *EQGAT* achieves speed-ups compared to models which base themselves off irreducible representations of the rotation group.

*Contributions*:

- *[C1]*: Usage and evaluation of an attention mechanism to propagate neighboring node features.
- *[C2]*: Usage of cross-product combinations of Cartesian vector features in the message construction.
- *[C3]*: Benchmark of the proposed *EQGAT* model on three atom3d tasks with (atom-level) graphs of size ~500--6'000 nodes (c.f. atom3d, fig 1: <https://arxiv.org/pdf/2012.04035.pdf>) and comparison against recent literature such as PaINN, GVP and SEGNN.

-------------------------------------------------------

**Strengths & Weaknesses**:

*Pros*:

- *[S1]* The paper tackles an interesting structural biology problem, for which equivariance to rotations plays an important role.
- *[S2]* The figures are well-chosen: They are helpful to facilitate an understanding of the paper and provide a good summary.
- *[S3]* The provided codebase for the submission is well-structured and readable. It executes properly (after renaming `moduXXXX-2s.py -> modules`).

*Cons*:

- *[W1]* It seems some key references are missing: The abstract mentions the introduction of a *novel attention mechanism*, but the attentive mechanism (Eq. 4 & 5) looks very similar to the GATv2 paper 'How attentive are graph attention networks?' (<https://arxiv.org/pdf/2105.14491.pdf>). Yet, *GATv2* (or the earlier *GAT* work <https://arxiv.org/pdf/1710.10903.pdf>) is not cited. It would be very helpful if the authors can point out in which way this attention mechanism is novel and how it compares to previous approaches.
- *[W2]* I am not yet convinced by the evaluation: The RMSE on the LBA task is the only score which outperform other equivariant GNN based methods without significantly overlapping error bars. For the PSR task: While *EQGAT* slightly outperforms *GVP-GNN* on the global $R_S$ metric, the difference is minimal and within the given variance. On the mean $R_S$ metric, it is well behind *GVP-GNN*. For the RSR task: The high variance of the RSR task -- which the authors also remark on in line 311 -- again makes it hard to draw a conclusive statement. While *EQGAT* performs on reasonably well overall, its variance again overlaps significantly with *GVP-GNN* and *PaiNN*. Taken together, this makes it hard to assess to which extent the proposed architecture in this paper (in particular its reliance on additive attention & cross-product combinations) are relevant to improve performance. An ablation study without these architectural features would be useful to shed light on the usefulness of these components. This would also allow the authors to quantitatively assess beliefs such as (line 333 onwards): "[...] we believe that including the cross product in our vector message allows the model to capture more of the geometric details [...]".
- *[W3]* Many of the modelling choices seem somewhat ad-hoc, because no explanation is given as to why something one option is chosen. For example, why did the authors choose to use an additive instead of a multiplicative attention mechanism? Why did the authors use attention-aggregation on scalar features, but normal gating (ie. without softmax) on vector features?

-------------------------------------------------------

**Recommendation**:
I vote for a weak rejection. The evaluation needs to be improved to make clearer statements. The results on the three provided atom3d benchmarks are not conclusively in favor of the proposed *EQGAT* method, and it is often only is marginally better than the competitors. This makes it hard to assess the usefulness of the proposed architecture. I suggest the authors to perform ablations of their suggested components and a more detailed discussion on why certain modelling choices were made (e.g. additive attention, attention gating of scalar features but not of vector features) to provide more clarity.

-------------------------------------------------------

**Questions**:
(See Cons above for more details)

- *[Q1]* Why did the authors choose to use an additive instead of a multiplicative attention mechanism?
- *[Q2]* Why did the authors use attention-aggregation on scalar features, but normal gating (ie. without softmax) on vector features?
- *[Q3]* For which types of structures is the cross-product feature relevant and for which not?

-------------------------------------------------------

**Additional feedback**:

Minor typos:

- Line 30: `a widespread of molecular structures`
- Line 234: `would result into a`

Minor presentation suggestions:

- I found section 4.3 hard to read. There were many indices which where not properly defined (e.g. $n$ in Eq. 6) and seem to go missing on one side of the equation. The sentences in the consecutive paragraph (e.g. line 181) are long and I found them hard to parse.

- Table 1 is cumbersome to assess and a simple two-sided confidence interval likely does not capture the distribution well. It would be great to see the distributions of the predictions to better assess and compare the methods. A violin plot is a simple way to do so. This should be done at least for the methods which were re-implemented. The methods for which values were taken from the literature could be plotted as mean with two-sided errorbars (as currently cited in the table) as well. This would make the information much easier to parse for readers.

-------------------------------------------------------

**Type of paper**: Full paper proceedings track submission.

-------------------------------------------------------
## Update after revisions:
I thank the authors for their extra work and for clarifying my questions and providing an additional ablation study. While many ideas in this paper have been around before in isolation, the revised paper now provides an interesting benchmark and ablation of using the cross product in low-tensor-rank geometric GNNs and of using multiplicative attention on updates on the invariants. I believe there is space for deeper studies, but I also believe this study is one that can benefit the community as a whole. Therefore, in recognition of the revisions the authors have made, I raise my score to an "accept".

---

### Meta-Review · Area_Chair_Nqfh · 2022-11-16

**Confidence:** 3
**Recommendation:** Accept

**Meta Review:**

Three out of four reviewers have converged on a score of 6 (weak accept), with reviewer 359T voting 5 (weak reject), making this a somewhat borderline paper. The main concerns that were raised:
1. The architectural contribution is somewhat incremental (359T, CRBR)
2. Limited evaluation / ablations / experimental results (aj5D, Chvz, 359T, CRBR)

Regarding 1: In my view the proposed architecture is indeed firmly within the space of architectures being considered in the literature, and bears similarities to other works. However, details are important and publishing them can be very helpful to those working on similar problems. This paper proposes an architecture that is well suited to the particular problem at hand, which involves larger geometric graphs than were considered in most related works. So I consider this a decent engineering contribution. After revision the paper also does a reasonable job at demonstrating which aspects are important for good performance.

Regarding 2: whereas the initial submission was lacking in terms of experimental validation of the method, the additional experiments were well received by the reviewers. The method achieves SOTA on RES and PPI, and an ablation study was added.

Overall my judgement is that the updated paper meets the bar.

---

### Decision · Program_Chairs · 2022-11-23

Accept (Poster)